# vcfdist: accurately benchmarking phased small variant calls in human genomes

Tim Dunn [1] ✉ & Satish Narayanasamy [1]

Accurately benchmarking small variant calling accuracy is critical for the continued improvement of human whole genome sequencing. In this work, we show that current variant calling evaluations are biased towards certain variant representations and may misrepresent the relative performance of different variant calling pipelines. We propose solutions, first exploring the affine gap parameter design space for complex variant representation and suggesting a standard. Next, we present our tool vcfdist and demonstrate the importance of enforcing local phasing for evaluation accuracy. We then introduce the notion of partial credit for mostly-correct calls and present an algorithm for clustering dependent variants. Lastly, we motivate using alignment distance metrics to supplement precision-recall curves for understanding variant calling performance. We evaluate the performance of 64 phased Truth Challenge V2 submissions and show that vcfdist improves measured insertion and deletion performance consistency across variant representations from $R^2 = 0.97243$ for baseline vcfeval to 0.99996 for vcfdist.

The first draft reference of the human genome was assembled with great difficulty in 2000 at an estimated cost of $300 million[1]. Following this massive effort, it became standard practice for whole genome sequencing (WGS) analyses to align sequencing reads to this reference to determine an individual's genome[2]. Because there is only around 0.1% difference in the genomic sequence of two individuals when excluding structural variants (SVs)[3,4], the final sequence is stored as a set of variations from the reference genome and reported in variant call format (VCF). These small germline variants (under 50 base pairs) are then classified as either single nucleotide polymorphisms (SNPs) or insertions/deletions (INDELs). Any variants which cannot be represented as a single SNP or INDEL are called "complex".

The past decade has been a time of rapid advancement in DNA sequencing chemistries, machine learning, and bioinformatics algorithms[5,6]. In such an environment, the ability to accurately compare the performance of variant calling pipelines is crucial. Firstly, accurate comparisons are necessary for identifying the current best-performing variant calling pipeline; this choice will impact real clinical decisions[7]. Secondly, researchers must be able to correctly identify promising avenues of further research. Thirdly, the curation of high-quality databases of germline mutations is only made possible with accurate comparison techniques. These databases are later used for linking genotypes with clinically relevant phenotypes[4,8].

The current standard benchmarking tool for variant calling is vcfeval, introduced by Real Time Genomics (RTG) in 2015[9] and later backed by the Global Alliance for Genomics and Health (GA4GH) Benchmarking Team[10]. It is now widely accepted as the standard tool for benchmarking small variant calls[10–13]. vcfeval uses an innovative pruned search algorithm to determine the largest possible matching subsets of query and truth variants. Regardless of the query and truth variant representations, vcfeval is able to determine equivalency as long as there is a complete exact match. A noteworthy extension is VarMatch, which uses a similar algorithm but implements several different optimization criteria and presents an algorithm to partition the inputs into smaller subproblems without a loss of accuracy[14].

Prior works have invested significant effort in defining one accepted representation for a given variant. The standard process of VCF variant normalization requires that variants are decomposed, trimmed, and left-shifted[14,15]; an example can be seen in Supplementary Fig. 1. Although normalization is sufficient to ensure there is one possible representation for a single variant, there is not a unique representation for multiple adjacent (complex) variants. This makes evaluating variant calling accuracy difficult[13].

---

[1]Computer Science and Engineering, University of Michigan, 2260 Hayward Street, Ann Arbor, MI 48109, USA. ✉e-mail: timdunn@umich.edu

It should be noted that competing standards for variant representation exist in other domains. For human-readable text-based variant descriptions (e.g. `NC_000001.2:g.120_123insAGC`), the Human Genome Variation Society (HGVS) standard[16] requires that variants be right-shifted. When representing variants and their associated metadata in databases, the SPDI[17] and VRS[18] data formats use the variant overprecision correction algorithm (VOCA). This algorithm specifies that when representing an INDEL that could be left or right-shifted, instead of arbitrarily placing this INDEL at the left or right end of this region, the entire region is reported with and without the INDEL. This strategy avoids an overly precise placement of an INDEL.

Despite its impressive advancement over previous state-of-the-art algorithms for variant comparison, vcfeval suffers from several key limitations which we attempt to address in this paper. Each limitation increases the extent to which vcfeval's results depend upon the initial representation of variants in the query and truth VCFs. Firstly, vcfeval has no notion of partial credit and will not report any true positives if no subsets of query variants match exactly with truth variants. Figure 1 demonstrates a simplified example of where different variant representations of the same sequence lead to significantly different measured accuracies. Refer to Supplementary Fig. 2 for a realistic example. Secondly, vcfeval outputs precision-recall curves for SNP and INDEL variants separately. Although precision and recall are concise metrics for understanding approximate variant calling accuracy, they are dependent on both the reference FASTA and query VCF representation. For complex variants, especially in low-complexity (repetitive) regions, the same query sequence can be represented in numerous equally valid ways with a differing total count of SNPs and INDELs (see Fig. 1a). Thirdly, vcfeval was designed to handle unphased query variants, and allows arbitrary local phasing of adjacent heterozygous variants. Allowing different phasings leads to different sets of possible query sequences depending on the original variant representation. While vcfeval does contain an experimental option for global phasing (`-Xobey-phase`), enforcing local phase is not currently supported[9].

In combination, these characteristics bias evaluations towards a particular variant representation that uses many SNPs rather than a few INDELs to represent genomic variation. This bias becomes most evident in high-quality variant callsets where most remaining variant calling errors occur in low-complexity or high-variance regions. As read sequencing technologies have recently improved in terms of both average read length and quality, we are now able to call variants in more difficult regions than ever before[19]. INDELs are orders of magnitude more likely to occur in low-complexity repetitive regions[20], and so it is important that variant calling evaluation is not biased against newer tools which are more likely to identify copy number variations. Lastly, vcfeval suffers from the inability to evaluate large clusters with many nearby variants due to its exponential complexity[9,14]. Because these benchmarking issues occur during VCF evaluation, they are broadly applicable to all sequencing technologies.

In this work we present vcfdist, an alignment-based small variant calling evaluator that standardizes query and truth VCF variants to a consistent representation, requires local phasing of both input VCFs, and gives partial credit to variant calls which are mostly (but not exactly) correct. We show that the SNP and INDEL precision-recall curves reported by vcfdist are stable across variant representations. Furthermore, we introduce alignment distance based metrics for evaluation which are entirely independent of variant representation, and only measure the distance between the final diploid truth and query sequences. We then introduce a variant clustering algorithm which reduces downstream computation while also discovering long-range variant dependencies. We evaluate all 64 submissions from the precisionFDA's 2020 variant calling Truth Challenge V2 using both vcfeval and vcfdist on two high-quality ground truth datasets: the NIST v4.2.1 WGS benchmark, and the Challenging Medically Relevant Genes (CMRG) dataset. We find that vcfdist improves measured SNP and INDEL performance consistency across variant representations from vcfeval's $R^2 = 0.14542$ and $0.97243$ to $R^2 = 0.99999$ and $0.99996$, respectively. We note that vcfeval's SNP $R^2$ value is only so poor (0.14542) because one of our selected representations (design point $A$)

(a)

| Reference | ACCCTTTTTTG | Query | ACCTTTG | Truth | ACCCTTTG |

**Query VCF Representation 1**

| POS | REF | ALT |
|-----|------|-----|
| 3 | CCTTT | C |

**Query VCF Representation 2**

| POS | REF | ALT |
|-----|------|-----|
| 1 | AC | A |
| 4 | CTTT | C |

**Truth VCF**

| POS | REF | ALT |
|-----|------|-----|
| 4 | CTTT | C |

(b)

**vcfeval Summary Statistics**

| | TP | FP | FN | PP | Precision | Recall | F1 | F1 Q-score |
|---|----|----|----|----|-----------|--------|----|-----------|
| Query Repr. 1 | 0 | 1 | 1 | 0 | 0.00 | 0.00 | 0.00 | 0.00 |
| Query Repr. 2 | 1 | 1 | 0 | 0 | 0.50 | 1.00 | 0.67 | 4.77 |

(c)

**vcfdist Summary Statistics**

| | TP | FP | FN | PP | Precision | Recall | F1 | F1 Q-score |
|---|----|----|----|----|-----------|--------|----|-----------|
| Query Repr. 1 | 0 | 0 | 0 | 1 | 0.67 | 0.67 | 0.67 | 4.77 |
| Query Repr. 2 | 1 | 1 | 0 | 0 | 0.50 | 1.00 | 0.67 | 4.77 |

(d)

**vcfdist Distance Summary**

| | ED | DE | DE Q-score | ED Q-score | ALN Q-Score |
|---|----|----|-----------|-----------|-------------|
| Reference | 3 | 1 | | | |
| Query Repr. 1 | 1 | 1 | 4.77 | 0.00 | 3.01 |
| Query Repr. 2 | 1 | 1 | 4.77 | 0.00 | 3.01 |

**Fig. 1 | A simple vcfdist partial credit example.** A simple example of vcfeval and vcfdist evaluations, demonstrating vcfeval's dependence on variant representation and the usefulness of partial credit. **a** Reference, query, and truth sequences, as well as the query and truth variant call files (VCFs). **b** vcfeval and **c** vcfdist count of true positive, false positive, false negative, and partial positive variants, as well as the calculated precision, recall, and F1 quality scores. Note that although both query VCF variant calls result in the exact same query sequence, the summary statistics differ. Partial credit alleviates this problem for vcfdist. **d** Distance-based summary statistics reported by vcfdist: edit distance, distinct edits, and alignment distance, which are independent of variant representation. An explanation of these summary statistics can be found in the Methods section.

never uses SNPs to represent variants. Excluding design point *A*, vcfeval's SNP $R^2$ value is 0.96888; however, we believe this design point is important to include in our analyses for reasons discussed later. In summary, vcfdist ensures consistent and accurate benchmarking of phased small variant calls regardless of the original variant representations.

## Results

### The affine gap design space for selecting variant representations

As demonstrated in Fig. 1, the main issue with a difference-based format such as VCF is that often there are multiple reasonable sets of variant calls that can be used to represent the same final sequence relative to a reference FASTA. Since DNA sequencing only measures the final sequence, there is no way of knowing which set of variants physically occurred. We can only select a representation which contains the most likely set of variants, based on the relative likelihoods of various mutations occurring.

This problem of variant representation can be viewed as a query-to-reference pairwise global alignment problem, and the path of the alignment with the minimum penalty score can be used to derive an edit path, representing the most likely set of variants. This approach was first explored in[21] and termed "Best Alignment Normalization". Here, we present a more thorough exploration of the design space for normalized variant representation. During alignment, allowed

operations include matching, substituting, inserting and deleting bases with corresponding penalties $m$, $x$, $g(n)$, and $g(n)$. Under the gap-affine model originally proposed in ref. 22, $g(n) = o + ne$, where $n$ is the length of the gap, $o$ is a gap-opening penalty, and $e$ is a gap-extension penalty. The relative value of substitution, insertion, and deletion penalties is critical, as it determines which variant representation is selected in the VCF.

Recent efforts such as refs. 23, 24 have demonstrated how to transform gap cost models into other equivalent representations. In Fig. 2a we normalize penalties such that $m = 0$ and then explore the design space for a general-purpose aligner with an affine-gap cost model. Black shaded areas represent the invalid areas where $o < 0$ or $e < 0$; opening and extending gaps should be penalized, not preferred over matching bases. We have also marked $2(o + e) < x$. Left of this line, a substitution will instead always be represented as a single-base insertion and a single-base deletion. Although this is clearly not ideal for a general-purpose aligner, it may make sense in repetitive regions of the genome where copy number variants are likely.

Figure 2a plots the default parameter configurations for the most commonly used aligners in the 2020 pFDA variant calling challenge. This plot also includes the default parameters for a wider range of tools, such as those used for structural variant (SV) detection (verkko, NGMLR)[25,26], copy number variant (CNV) detection

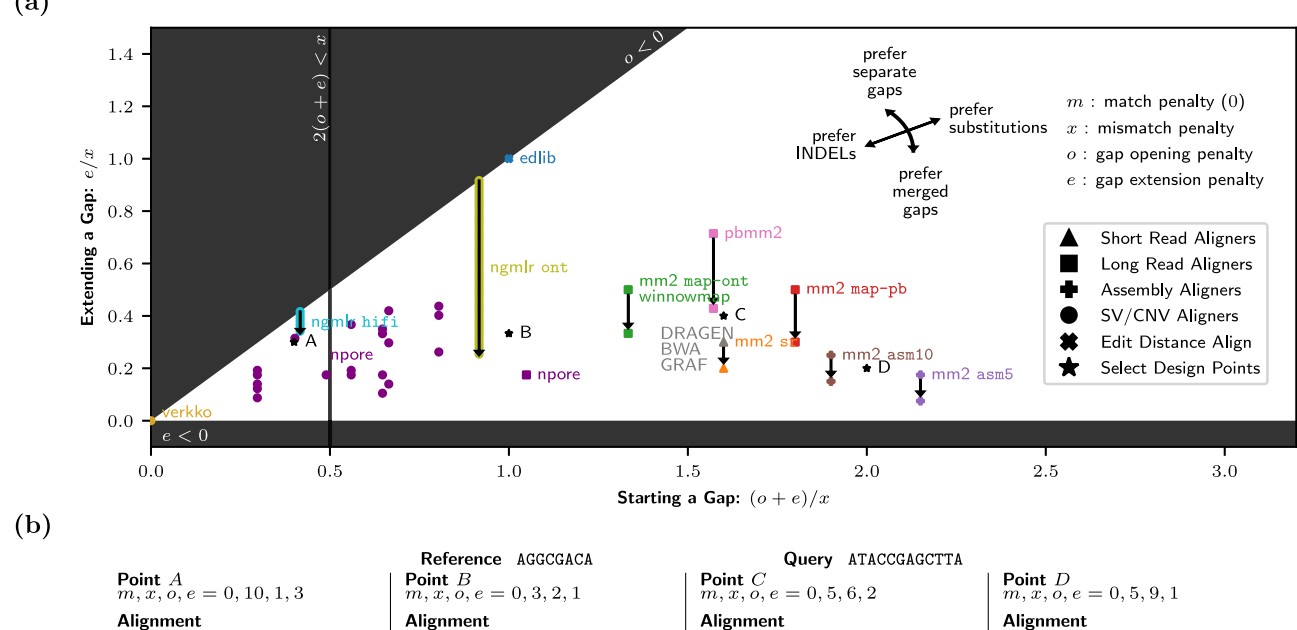

**(a)**

**(b)**

**Fig. 2 | The affine gap design space for alignment and variant representation. a** The design space for an affine-gap aligner with match, mismatch, gap opening, and gap extension penalties $m$, $x$, $o$, and $e$. All parameters have been normalized so that $m = 0$ (see Supplementary Fig. 3 for details), and the penalties for starting ($o + e$) and extending ($e$) a gap are plotted relative to substitutions ($x$). This plot includes the aligners used in the precisionFDA's Truth Challenge V2, as well as assembly, edit distance, and copy number/structural variant (CNV/SV) aligners for comparison. Each aligner is plotted in a unique color, except for when multiple aligners use identical parameters. For dual affine gap aligners, two points are

plotted with an arrow indicating the transition to a lower extension penalty $e_2$. NGMLR[26] uses a logarithmic gap penalty, and so there is a continuous lowering of $e$. nPoRe[20] uses different gap penalties for simple tandem repeats (STRs) based on their measured likelihoods, resulting in many plotted points. **b** Aligning the same query and reference sequence using different gap parameters (design points *A*, *B*, *C*, and *D*) results in different sets of reported variants. Each variant call file (VCF) shows the variant positions (POS) in addition to the reference (REF) and alternate (ALT) alleles.

(nPoRe)[20], edit distance calculation (edlib)[27], and assembly alignment (minimap2's `asm5` and `asm10` configurations)[28]. The original and normalized affine-gap parameters for each tool configuration are included in Supplementary Fig. 3.

The various configurations for minimap2 are depicted with two data points and a transition, since minimap2 uses double affine gap alignment[28]. Under double affine gap alignment, there are two separate gap-opening ($o_1$, $o_2$) and gap-extension ($e_1$, $e_2$) penalties, and the minimum of $g_1(n) = o_1 + ne_1$ and $g_2(n) = o_2 + ne_2$ is preferred. Typically, $o_2 > o_1$ and $e_2 < e_1$. As a result, the penalties will be equal at $n = \frac{o_2 - o_1}{e_1 - e_2}$; prior to this point, $g_1 < g_2$ and afterwards $g_2 < g_1$. Thus $g_1$ determines the penalty for a short gap and $g_2$ determines the penalty for a long gap.

Any tools using homopolymer (HP) compression or simple tandem repeat (STR) compression such as verkko[25] are plotted at (0, 0) in Fig. 2a, since compressed gaps are not penalized in repetitive regions during alignment. Repetitive sequence compression is common practice for long-read de novo assemblers[25,29,30], since incorrect estimation of homopolymer run length is the dominant error mode of nanopore sequencing[31]. nPoRe penalizes copy number variation in these repetitive regions only slightly, depending on the copy number and repeat unit length[20]. This results in many different points on the left side of Fig. 2a. NGMLR, an aligner for structural variant detection, also does not heavily penalize INDELs[26]. In contrast to minimap2, NGMLR uses a smoother convex gap cost model, decreasing the penalty for each additional gap extension as the length of the gap grows. As a result, NGMLR's gap model is represented in Fig. 2a as a smooth gap extension penalty decrease.

In Fig. 2a, design point $A$ was selected as the approximate centroid of the CNV/SV aligners. Likewise, point $C$ was selected as the approximate centroid of common short and long read alignment parameter configurations, and point $D$ was selected for assembly aligners. Lastly, point $B$ was selected because this design point will simultaneously minimize the edit distance (ED) and number of distinct edits (DE) in the variant representation, weighting DE twice as heavily as ED. It is also the approximate midpoint of design points $A$ and $C$. All selected points were additionally chosen such that their substitution and gap penalties can be represented by small positive integer constants (see Fig. 2b).

### A standard complex variant representation: design point C
As Fig. 2b shows, for complex variants there are many reasonable representations which depend upon the selected affine gap alignment parameters. By varying alignment penalties $m$, $x$, $o$, and $e$, Fig. 2b shows that aligning the same query and reference sequence will result in a different VCF representation at each of the four selected design points $A$, $B$, $C$, and $D$. If we define a standard set of affine gap parameters, however, then there is usually just one possible representation for complex variants such as this.

We propose using design point $C$, where $(m, x, o, e) = (0, 5, 6, 2)$, as the standard set of affine gap parameters for reporting variants. When normalized to $m = 0$ and $x = 1$, $C = (0, 1, 1.2, 0.4)$. Design point $C$ was selected because it is the approximate centroid of widely accepted parameters used to align reads of three different popular sequencing technologies (Illumina short reads, PacBio HiFi reads, and Oxford Nanopore long reads). For Illumina short reads, DRAGEN, BWA, GRAF, and minimap2 `sr` use the same normalized alignment parameters (0, 1, 1.3, 0.3) (Supplementary Fig. 3). For ONT long reads, minimap2 `map-ont` and winnowmap use (0, 1, 0.833, 0.5). For PacBio HiFi reads, minimap2 `map-pb` uses (0, 1, 1.3, 0.5) and pbmm2 uses (0, 1, 0.857, 0.714). Of the 53 submission methodologies documented from the pFDA Truth Challenge V2, 52 aligned reads using one of these above-mentioned gap parameter configurations, all of which are relatively similar. The other aligner used was NovoAlign, whose relative alignment penalties depend upon base quality[32].

### INDEL/SNP precision and recall depend on variant representation
Variant calling accuracy is currently evaluated by measuring separately the SNP and INDEL precision-recall curves, and then reporting the precision and recall where the F1 score is maximized[9–12]. This metric is useful because it gives an intuitive overview of what percentage of variants were called correctly and incorrectly for SNPs and INDELs separately. Figure 3a, b, however, shows that precision-recall curves can change significantly depending on variant representation, even when the evaluated query and truth sequences are the exact same. This is partially because the total number of SNP and INDEL variants reported in a VCF will vary depending on the selected variant representation.

This issue is highlighted in Fig. 3b by design point $A$, which represents all SNPs as a 1-base insertion and 1-base deletion (because $2(o + e) < x$; see Fig. 2a). Since VCFs represented using point $A$ report no SNPs, any false positives will be categorized as INDEL FPs. True positive variants will be SNP TPs when the truth VCF contains a SNP, and INDEL TPs when the truth VCF contains an INDEL. This results in query VCFs at point $A$ having perfect SNP precision, but a lower INDEL precision than other representations. Similar problems exist for other variant representations, although to a lesser extent. The stabilized precision-recall curves following vcfdist's variant standardization can be seen in Figs. 3c, 4d.

### INDELs are common in repetitive regions: design point A
Demonstrating that standardizing query VCFs using design point $A$ changes the resulting evaluation may initially appear to be a contrived argument, since point $A$ is not similar to any aligners used in the pFDA Truth Challenge V2. Figure 3a shows little difference in the performance of VCFs represented using common aligner parameters such as BWA, minimap2 `map-ont`, minimap2 `map-pb`, and pbmm2[28,33,34]. This is because all of these general-purpose read aligners were designed for aligning reads in non-repetitive regions, to which most bioinformatics analyses have been restricted until recent years[35,36]. As such, their parameters are clustered in a small region of the affine gap penalty design space in Fig. 2a. We selected design point $A$ precisely because of its low gap penalties model how alignment occurs in repetitive regions by dedicated copy number variant (CNV) aligners or any aligners performing homopolymer (HP) or simple tandem repeat (STR) compression.

As the field shifts towards true whole-genome evaluation, we expect variant callsets to frequently include merged results from general-purpose aligners/callers in non-repetitive regions and CNV/SV aligners/callers in more repetitive regions. In low-complexity regions, CNVs and other small gaps are orders of magnitude more likely than elsewhere[20], necessitating the inclusion of variants whose representation falls on the left-hand side of Fig. 2a near design point $A$. As can be seen from Fig. 2a, several existing SV and CNV callers already occupy this space. When these repetitive INDEL callsets are merged with callsets from non-repetitive regions, we must ensure that our evaluation of variant calling accuracy remains unbiased.

### Correct local phasing of dependent variants is critical
Incorrect local phasing of heterozygous variants leads to an entirely different sequence of bases for both haplotypes, which may impact clinical decisions[37]. Despite the relative ease of local phasing given new long-read sequencing technologies, vcfeval does not enforce that evaluated VCFs contain phasing information. In fact, vcfeval discards this information when available, considering any possible local phasing to be correct[9]. To illustrate why this is a problem, the original query VCF representation shown in Fig. 3d unnecessarily fragments a single heterozygous variant into five heterozygous variants. For these 5 variants, there are now $2^5 = 32$ possible local phasings, each resulting in a different pair of sequences. Perhaps unsurprisingly, one of these 32 phasings (and not the phasing initially reported) results in a match with

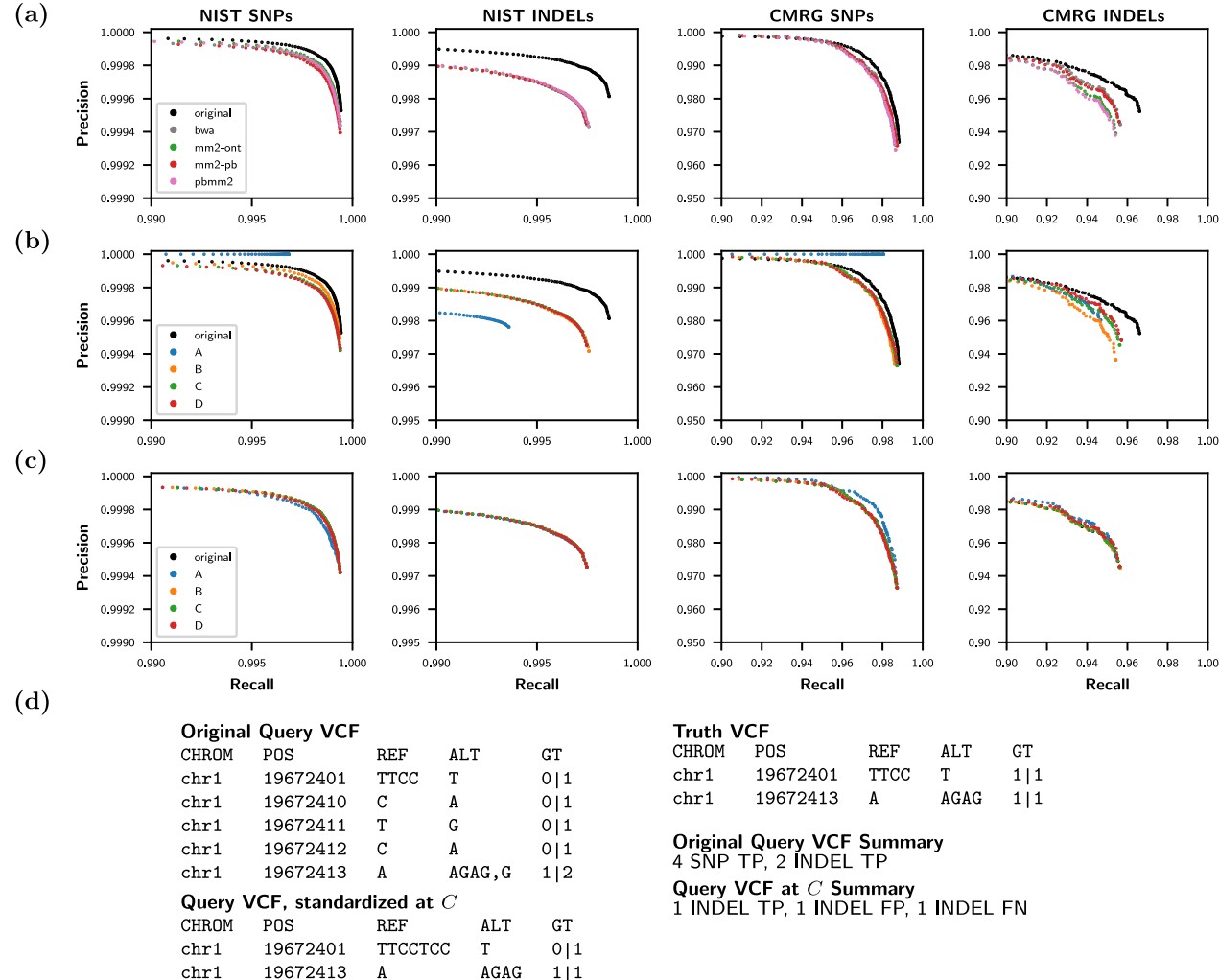

**Fig. 3 | vcfeval baseline precision and recall.** vcfeval precision-recall plots for Truth Challenge V2 submission `K4GT3` on the NIST whole genome and Challenging Medically Relevant Genes (CMRG) datasets for single nucleotide polymorphisms (SNPs) and insertions/deletions (INDELs) separately. **a** Evaluating the original query variant call file (VCF) and after changing the query variant representation using the alignment parameters of common aligners (see Fig. 2). **b** Evaluating the original query VCF and after changing the query variant representation to design points *A*, *B*, *C*, and *D* (see Fig. 2). **c** Standardizing the five representations from (b) using vcfdist prior to evaluating with vcfeval improves consistency. **d** A real example

demonstrating why the original `K4GT3` query VCF appears to significantly outperform other representations in (**a**) and (**b**). Each VCF shows the variant chromosomes (`CHROM`) and positions (`POS`) in addition to the reference (`REF`) and alternate (`ALT`) alleles and their genotypes (`GT`). Because vcfeval discards query phasing information and allows any possible local phasing, the original fractured variant representation is considered entirely correct (all true positives) whereas the more succinct standardized representation at *C* is not (it contains false positives and false negatives). Source data are provided as a Source Data file.

the ground truth, and all variants are considered true positives. This serves to heavily bias evaluations towards fragmented query variant representations.

In contrast, vcfdist requires locally phased truth and query input VCFs, and requires local phasing to be correct in order to consider variants true positives. This results in a lower precision-recall curve in Fig. 4d than Fig. 4a, which we believe more accurately represents the true performance. It also makes evaluation of the original fragmented representation (black) more consistent with the other representations (Fig. 4b vs Fig. 3b). Importantly, vcfdist does not require correct global phasing of variants, and allows for arbitrary switch errors to occur between clusters of dependent variants. Figure 5 shows an overview of our tool, vcfdist.

**A standard variant representation stabilizes precision-recall curves**

The precision-recall curve stabilization gained by using a standard variant representation (point *C*) is clearly demonstrated in

Figs. 3, 4. In fact, standardizing variant representation is sufficient for prior work vcfeval to obtain consistent results across variant representations (Fig. 3c). Researchers could use vcfdist as a preprocessing step for variant standardization, and then perform evaluations as usual through vcfeval. It is important to note, however, that consistency does not imply accuracy. The precision-recall curve for vcfdist in Fig. 4d is not the same curve as for vcfeval in Fig. 4a. The other improvements presented in this paper in regards to clustering, phasing, and partial credit have a large impact on the final results.

As can be seen in Fig. 6a, b, the final F1 scores reported on both the NIST and CMRG datasets are more consistent for vcfdist than vcfeval as variant representation changes. For each graph, the coefficient of determination ($R^2$) is greater and the AMRC is lower for vcfdist in comparison to vcfeval. We define AMRC (Average Maximum Rank Change) in the Methods section. It is an average measure of how much a particular submission's rank (in performance relative to the other 63 submissions) would change when using the best versus the worst

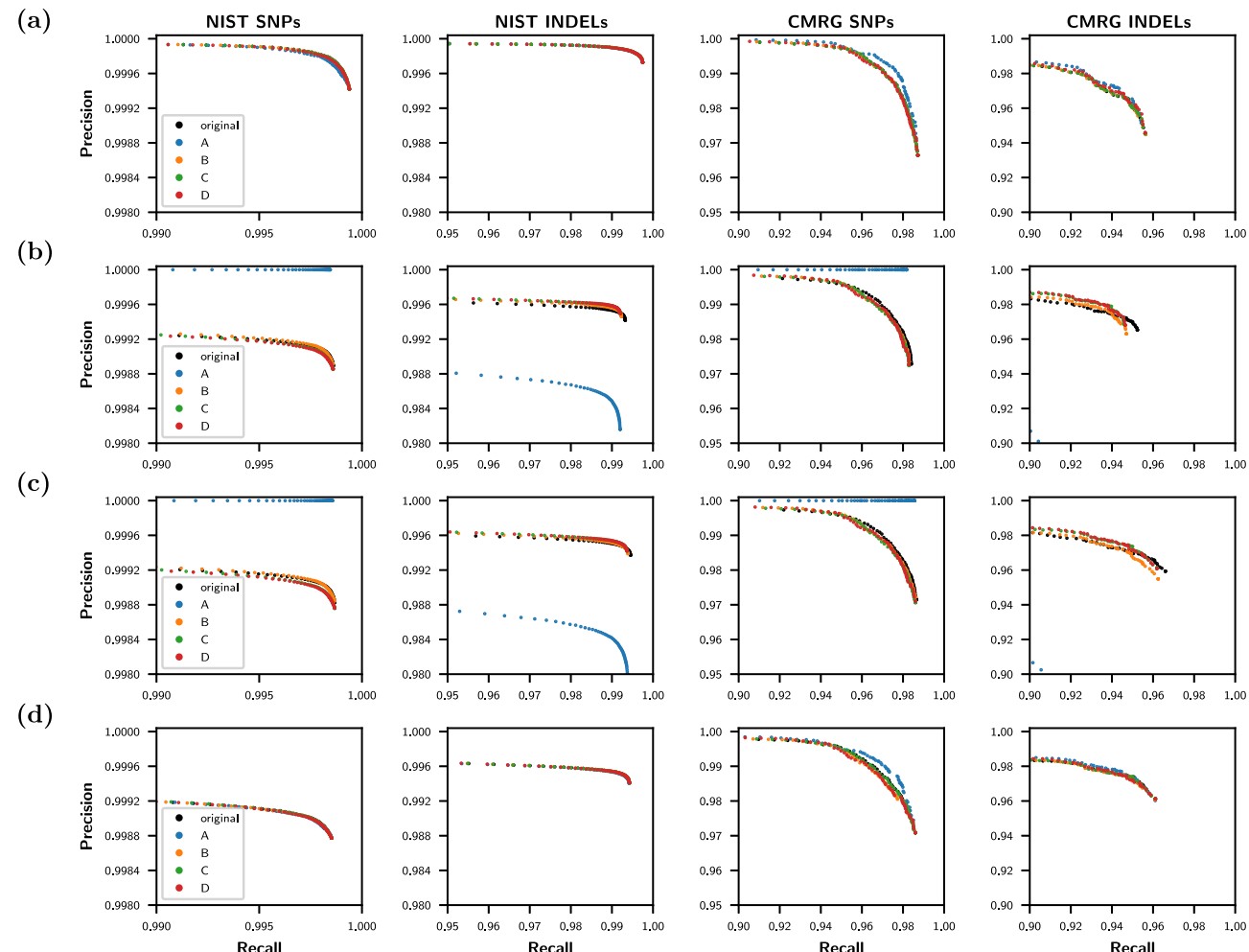

**Fig. 4 | vcfdist precision and recall. a** Standardized vcfeval and (**b**–**d**) vcfdist precision-recall plots for Truth Challenge V2 submission `K4GT3` on the NIST whole genome and Challenging Medically Relevant Genes (CMRG) datasets, for single nucleotide polymorphisms (SNPs) and insertions/deletions (INDELs) separately. For all plots, the original query VCF is evaluated before and after changing its variant representation to design points *A*, *B*, *C*, and *D* (see Fig. 2). **a** This plot is identical to Fig. 3c, but using axes consistent with the remainder of this figure. **b** Evaluation with vcfdist, turning the options for standardization and partial credit off; this can be directly compared to vcfeval's performance in Fig. 3b. Note that the original representation no longer outperforms other representations, since local phasing is enforced. **c** Evaluation with vcfdist, allowing partial credit but not standardization, resulting in minor CMRG recall improvements. **d** Evaluation with vcfdist, allowing partial credit and standardizing variant representation. This results in the most consistent results, and is the recommended usage. Source data are provided as a Source Data file.

possible variant representation for that particular submission. Ideally, values of $R^2 = 1$ and AMRC = 0 would indicate that our evaluation is entirely independent of variant representation.

### Partial credit more accurately reflects variant calling performance

As demonstrated in Fig. 1, variant calls which are nearly but not exactly correct are penalized under current evaluation methods. Figure 4c shows a slight improvement in variant recall over Fig. 4b for the CMRG dataset after vcfdist assigns partial credit to mostly correct variants.

Importantly, we note that whether or not to use partial credit when calculating summary metrics depends on the application. For general variant call benchmarking, we recommend partial credit because INDELs less than 50 base pairs can be called mostly correct and partial credit better reflects the variant calling performance. Partial credit may also be useful when calling STR repeat lengths. It should not, however, be used for clinical evaluation of small INDELs in regions of coding DNA where a slightly incorrect call will result in a frame shift mutation.

### Distance-based performance metrics provide additional insight

As mentioned earlier, precision-recall curves are dependent upon query and truth VCF variant representations, as well as the reference FASTA. In contrast, distance-based performance metrics such as the edit distance, the number of distinct edits, and alignment score are determined solely by the query and truth sequences and are unchanged by different variant representations or references. Figure 6c demonstrates this. The minor remaining differences in distance-based metrics are caused by false-positive variant dependencies during clustering. Since slightly different clusters are created, depending on the original variant representations, enforcing consistent local phasing within a cluster may lead to different results.

Although a similar level of evaluation consistency can be gained by using a standard variant representation with vcfdist (Fig. 6b), distance-based summary metrics are useful to gain a better understanding of overall performance. Supplementary Fig. 5b shows that of the three multi-technology pFDA submissions which tied for the winning performance on the NIST dataset, Sentieon's solution performed best on SNPs, but Roche's

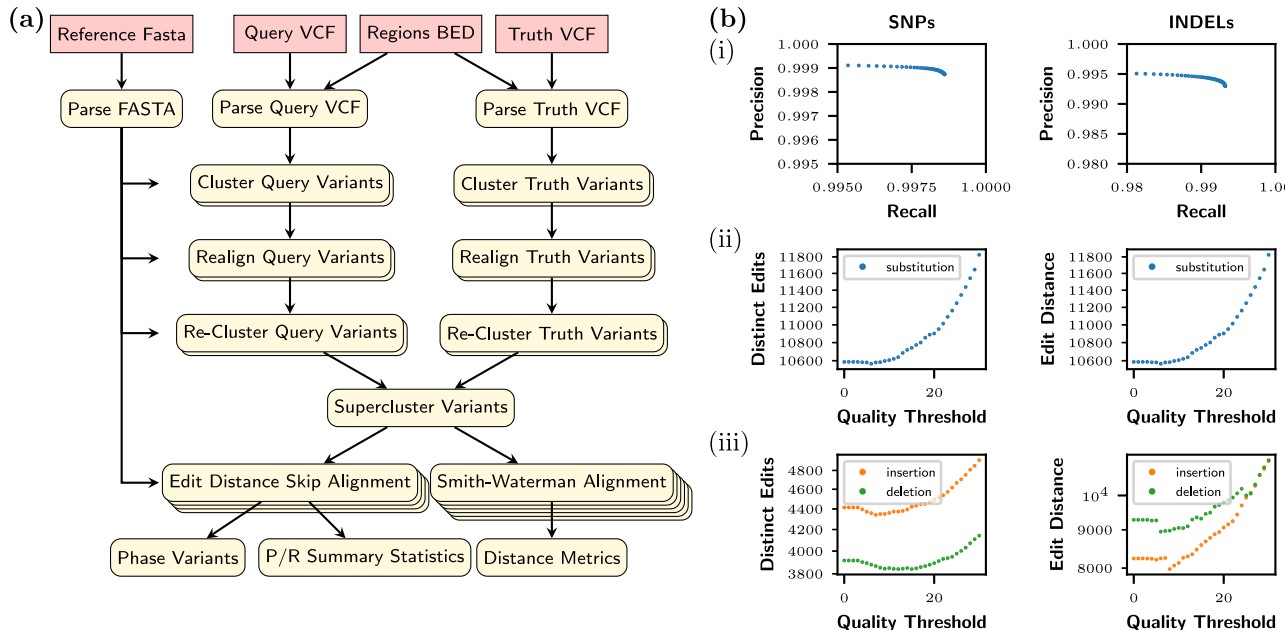

**Fig. 5 | Overview of vcfdist. a** Diagram of vcfdist workflow; more details regarding each step can be found in the Methods section. **b** Visual summary of vcfdist results for submission K4GT3; a textual summary is given in Supplementary Fig. 4. (i) Precision-Recall curve (ii) Single nucleotide polymorphism (SNP) and (iii) Insertion/deletion (INDEL) edit distance (ED) and distinct edit (DE) error curves, sweeping possible quality thresholds. Source data are provided as a Source Data file.

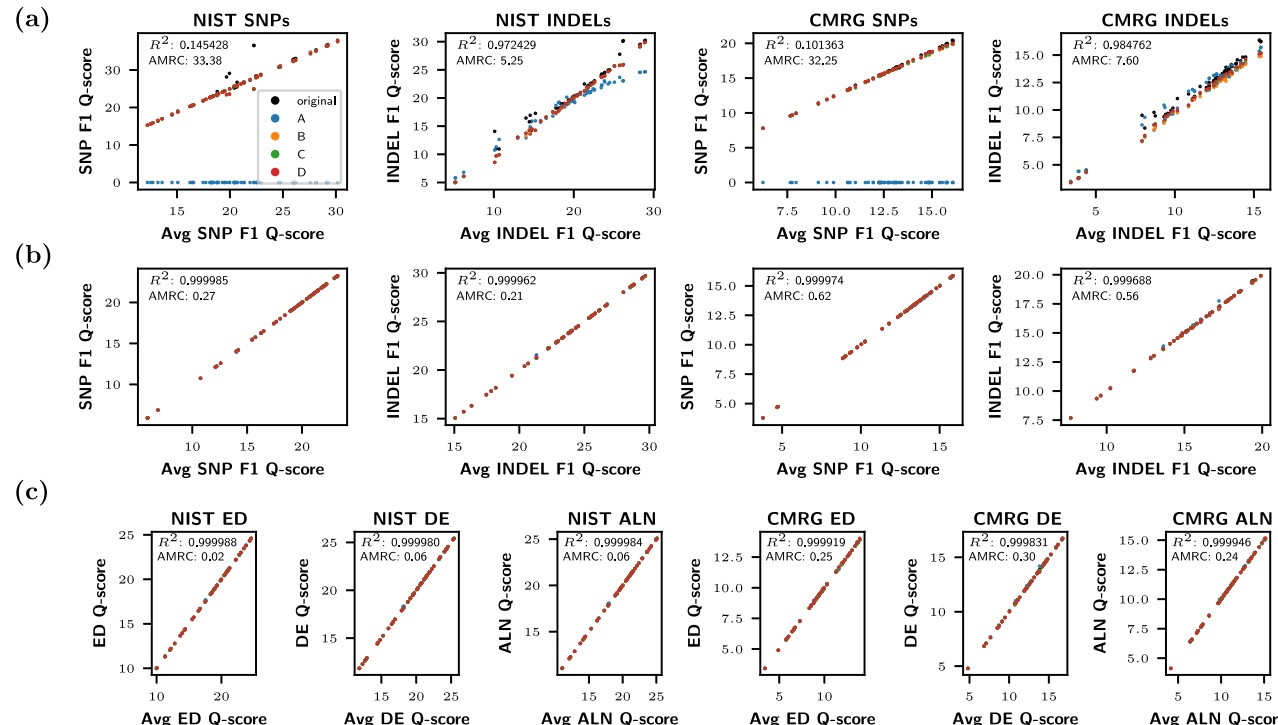

**Fig. 6 | Evaluation stability results for vcfeval and vcfdist. a** vcfeval and **b** vcfdist F1 Q-score plots for all Truth Challenge V2 submissions on both the NIST whole genome and Challenging Medically Relevant Genes (CMRG) datasets, for single nucleotide polymorphisms (SNPs) and insertions/deletions (INDELs) separately. **c** The same information, using sequence distance-based summary metrics (edit distance, distinct edits, and alignment distance; see the Methods section for a full explanation of each). On each graph, average Q-score is plotted against the Q-score for the original representation and at points *A*, *B*, *C*, and *D* (see Fig. 2). Clearly, vcfdist results in (**b**) and (**c**) are more stable in regards to variant representation than vcfeval in (a). Source data are provided as a Source Data file.

solution performed best on INDELs. Using precision-recall curves for SNPs and INDELs separately, it is hard to get a clear picture of which submission has the best overall performance. This is because SNPs are much more common than INDELs, and the two variant types do not always have similar accuracy characteristics

(performance on INDELs is usually worse). Supplementary Fig. 5c clearly demonstrates that Roche's solution performs better overall on the NIST dataset, as it has lower edit distance, distinct edits, and alignment distance. Interestingly, Sentieon's solution achieves better performance on the CMRG dataset.

**Computational phasing errors do not impact our conclusions**

Because the majority of pFDA VCF submissions were unphased, we employed computational phasing (see Methods) in order to evaluate vcfdist on phased VCFs. There are two possible types of phasing errors that could arise: flip errors (in which a single variant is assigned the incorrect haplotype) and switch errors (in which many adjacent variants are assigned to the incorrect haplotype).

vcfdist is able to measure the switch error rate directly using the phasing analysis algorithm depicted in Supplementary Fig. 9. vcfdist detected 1337 switch errors for submission K4GT3, evaluated on the NIST dataset, which includes 5,423,813 variants. This amounts to one switch error every 4,057 variants (0.025%), or every 2.3 Mb. Unfortunately, vcfdist is unable to directly measure the flip error rate. However, we can use the accuracy of the best-performing submissions to get an upper bound. Since the maximum SNP F1 score measured by vcfdist on the top few pFDA submissions is above Q35 (99.97% accuracy), the phasing error rate of individual variants is at most 0.03%.

Because vcfdist only enforces local and not global phasing, switch errors are allowed to occur between superclusters without affecting vcfdist's analysis whatsoever. Switch errors are measured and reported later, but do not affect the total counts of true positive, false positive, and false negative variants.

Although flip errors impact the measured accuracy, they do not meaningfully impact the conclusion that vcfdist is less sensitive to variant representation than vcfeval as a result of our methodological contributions (Fig. 6). Since phasing occurs upstream to evaluation with vcfdist, flip errors could reduce the measured performance of each variant caller (with the same impact as if the original variant caller had phased variants incorrectly), but this does not impact any of our results in regard to evaluation stability. In other words, flip errors will slightly lower the values of the precision-recall curves in Fig. 4d for each representation uniformly. The curves will remain similarly clustered, much closer together than when evaluated with vcfeval (Fig. 3b).

**Runtime and efficiency**

We evaluated the runtime of submission K4GT3 on the CMRG and NIST datasets for each of the five representations in Fig. 4 (original, A, B, C, D). On the relatively small CMRG dataset, the baseline vcfeval's single-threaded runtime was 153.6s. In comparison, vcfdist's average single-threaded runtime was 441.2s and its maximum memory usage was 4.99G. If, however, vcfdist skips VCF realignment and uses the simpler gap-based clustering method, its average runtime decreases to 25.9s and RAM usage is only 3.27GB. For smaller datasets, loading the reference FASTA accounts for the vast majority of memory usage (≈3GB).

On the much larger NIST dataset, vcfdist's maximum RAM usage only increases to 5.86GB. Its single-threaded runtime (2648.2s) remains approximately 2 × slower than vcfeval's (1489.6s), unless realignment and Smith–Waterman clustering are disabled (663.8s). The baseline, vcfeval, additionally supports multi-threading with contig-level granularity. We plan to add multi-threading support to vcfdist with supercluster granularity (allowing core usage to scale beyond 23 for human genomes) once development of other features has ceased. Since other steps in the whole genome sequencing pipeline are the computational bottleneck (basecalling, alignment, and variant calling)[12,20], this has not yet been a priority.

# Discussion

The first major step towards affordable whole genome sequencing was the development of second-generation sequencing methods, which enabled massively parallel sequencing of short reads of 100–1000 bases[38]. With such short reads, resolving phasing and calling variants in repetitive genomic regions was incredibly difficult. It was in this context that vcfeval was developed. With the introduction of third-generation sequencing methods in recent years, however, the scope of genomic regions under active analysis has expanded. The 2019 NIST v4.2.1 benchmarking dataset used in this paper includes 17% more SNVs, 176% more INDELs and 12% larger benchmarking regions than the 2016 NIST v3.3.2 dataset[39]. Additionally, now that datasets often have average read lengths of 10-100Kb, some recent genome assemblies contain phase block N50s of more than 20Mb[40]. As a result, local phasing information is highly accurate.

As the state of the field and technology shifts, so too must the standards for evaluation. The accuracy of third-generation sequencing has improved tremendously in the past few years[5,6], and most small variant calling errors are now limited to low-complexity regions and complex variants. This work aims to ensure that evaluation is consistent in low-complexity regions and for complex variants where many representations are possible. We do so by proposing a standard variant representation, enforcing consistent local phasing, improving variant clustering, attributing partial credit, and motivating the use of alignment distance metrics for evaluation.

Despite the widespread availability of accurate phasing information in research-based variant calling pipelines that use the latest long-read sequencing technologies, it would be remiss to ignore the fact that the bulk of sequencing that happens today is short-read based. For clinical applications, short-read sequencing is preferred due to the stability of the technology, its well-understood performance characteristics, and lower cost[41]. Nevertheless, we believe that long-read sequencing will ultimately dominate clinical practice due to its unique ability to phase variants, detect larger INDELs and SVs, and resolve repetitive or repeated genomic regions[12,13] We additionally expect the cost to lower as the technology matures. Because we intend for vcfdist to be used in evaluating long-read pipelines in the research and development phase, we have intentionally designed vcfdist so that phasing information is required.

As genome sequencing becomes cheaper and more accurate, and as reads lengthen, a gradual shift is also beginning from reference-based variant calling to de-novo diploid assembly[25,40]. Even if the most common format for storing a human genome progresses from a haploid reference FASTA and an accompanying VCF to a diploid FASTA, there will always be the need to compare human genomes to one another. Whether this comparison is stored using a VCF or graph-based format, the same challenge of multiple possible alignments remains. As a result, we expect this work to remain relevant for the foreseeable future.

# Methods

### Datasets

All datasets used in this manuscript have been made publicly available in prior work, and were for human genome HG002 using reference genome assembly GRCh38. This includes precisionFDA Truth Challenge V2 variant callsets[13], in addition to the NIST v4.2.1[19] and CMRG v1.00[35] ground truth callsets from the Genome in a Bottle Consortium (GIAB). For the NIST v4.2.1 dataset, we used the phased truth VCF `HG002_GRCh38_1_22_v4.2.1_benchmark_hifiasm_v11_phase-transfer.vcf.gz` instead of the unphased `HG002_GRCh38_1_22_v4.2.1_benchmark.vcf.gz` VCF. For more information, please refer to the Data Availability section at the end of this manuscript.

### Preprocessing

Running our pipeline required two modifications to the phased NIST ground truth VCF. The CMRG ground truth VCF was used unchanged. Firstly, there were erroneously two additional blank FORMAT field values in the Major Histocompatibility Complex (MHC) region on chr6 that were removed. Secondly, WhatsHap v1.7 only accepts integer phase set values, whereas the NIST truth VCF contained phase sets named with strings. The phase set (PS) tags in the NIST ground truth VCF were removed using bcftools annotate (samtools v1.16.1)[42].

This does not affect our downstream evaluation because vcfdist only assumes variants are phased locally, and not globally.

Several of the pFDA Truth Challenge V2 query VCFs required modification as well. Submissions 0GOOR and BARQS reported some reference or ungenotyped variants as haploid, which were filtered. Chromosomes were renamed (i.e. 1 to chr1) for the same two query VCFs. Submissions 0GOOR, BARQS, and HB8P3 reported variants on decoy contigs, which were filtered. Additionally, the VCF header for HB8P3 did not store the lengths of each decoy contig. Submission Y8QR8 did not report any contig lengths, which were added manually. Submissions WGQ43 and YGOTK did not contain variant quality scores. Submissions WX8VK and CZA1Y contained an erroneous variant where ALT was the same as REF at position chr6:28,719,765. This was removed.

### Phasing input query and truth callsets

vcfdist requires input query and truth VCFs to be locally phased. However, most VCFs submitted to precisionFDA's Truth Challenge V2 were unphased. In order to phase these VCFs prior to evaluation, we downloaded all the original HG002 sequencing read FASTAs provided to participants of the competition. This dataset contained 35 × genome coverage of Illumina short paired-end reads, 35 × coverage of PacBio HiFi reads, and 50 × coverage of Oxford Nanopore reads, sequenced on a PromethION and basecalled with Guppy v3.6.0. All reads were aligned to GRCh38 with minimap2 v2.24-r1122 using the default parameters corresponding to each sequencing technology (sr, map-pb, and map-ont)[28]. Supplementary and secondary alignments were filtered. The aligned reads were then phased using whatshap haplotag v1.7[43]. Lastly, all variants in the 64 Truth Challenge V2 submission callsets were phased using whatshap phase v1.7 and the phased reads from all three technologies.

Although the CMRG truth VCF was entirely phased, only 99.3% of variants in the NIST truth VCF were phased. For this work, we considered all truth variants to be phased even when they were not. Importantly, although this impacts the measured absolute performance of each query VCF, it does not impact any conclusions this study makes regarding evaluation stability. Due to recent improvements in read lengths and accuracies, most future benchmarking truth VCFs will likely be fully phased.

### Clustering dependent variants

When multiple variants occur at the same locus, there are often multiple ways that differences between the draft reference FASTA and the resulting sequence could be represented in a VCF. This is still true after performing standard variant normalization techniques. For an example, see Fig. 2b. In order to fairly evaluate the differences between two VCFs, we must evaluate an entire set of nearby variants (or "cluster") in such regions simultaneously. VarMatch proposes a clustering algorithm that can be used to identify separator regions (long and non-repetitive sections of the reference genome that contain no variants), which can be used to divide variants into independent clusters[14].

Using a similar approach, we note that two clusters of variants can be considered independent of one another for a given set of affine-gap parameters if all optimal alignment paths return to the main diagonal between the two clusters. This "main diagonal" is the reference region between the two clusters which contains no variants. In order to determine whether two adjacent clusters are dependent, we begin extending from the start of the first cluster and note the rightmost reference position to which we can extend while avoiding the main diagonal and having an alignment penalty of lesser or equal value to that of the original representation of variants within the first cluster.

We mirror this process for the second cluster, extending from its end to the leftmost reference position possible while avoiding the main diagonal and limiting the alignment penalty. If the latter position is less than or equal to the first, then these two paths are able to meet off the main diagonal, and these two clusters are dependent. We use an iterative algorithm, initializing each cluster to a single variant. If two adjacent clusters are dependent, we merge them into a single larger cluster. We repeat this process until all clusters have not grown from the previous iteration or the maximum iteration count is reached. An overview of this clustering algorithm, which was also partly inspired by the uni- and bi-directional Wave Front Alignment (WFA) algorithms[44,45], can be seen in Supplementary Fig. 6.

### A standard variant representation

As shown in Fig. 2a, point $C$ was selected to be the Smith-Waterman design point for a standard variant representation, since it is the approximate centroid of widely accepted parameters used to align reads of three different popular sequencing technologies (Illumina short reads, PacBio HiFi reads, and Oxford Nanopore long reads). The integer affine-gap parameters corresponding to point $C$ are $(m, x, o, e) = (0, 5, 6, 2)$. Each haplotype was standardized independently, for both the truth and query VCFs. After clustering variants, within each cluster all variants are applied to the draft reference FASTA and the resulting sequence is realigned using the standard affine-gap parameters for point $C$. A new set of variant calls is then defined from the resulting alignment string, now in the standard representation.

### Superclustering variants

After clustering and realigning the truth and query variants separately, we need some means of associating query variants with their corresponding truth variants and vice-versa. In order to do so, we group together clusters on both the truth and query haplotypes that reach within 50 base pairs of one another. An example is shown in Supplementary Fig. 7. This resulting "supercluster" may be composed of several variant clusters on any of the four haplotypes (two each for the variant and query). All subsequent analyses take place within a supercluster, which are considered fully independent of one another.

### Alignment and local phasing

Since vcfdist assumes variants have been locally phased, we do not consider the possibility of genotype or switch errors within a supercluster. However, because vcfdist makes no assumptions regarding global phasing of the truth and query VCFs, we consider the relative phasing of each supercluster to be unknown and allow phase switches in between superclusters. In order to phase superclusters, we first perform each of the four possible alignments of truth and query haplotypes: Truth1 to Query1, Truth1 to Query2, Truth2 to Query1, and Truth2 to Query2.

During this alignment, we represent each query haplotype as a graph (merged with the reference sequence) to allow skipping query variants without penalty, since there may be false positive variant calls. Note that although this algorithm is similar to ordinary sequence-to-graph alignment, it is not equivalent since reference-to-query transitions (and vice versa) are not allowed within truth variants. An example of this alignment is shown in Supplementary Fig. 8. The minimum edit distance is calculated for each of the four alignments, and the phasing which minimizes total edit distance is selected: (X) Truth1 to Query1 and Truth2 to Query2, or (Y) Truth1 to Query2 and Truth2 to Query1. This represents the best possible phasing for the supercluster.

### Partial credit

Once a phasing has been selected, summary statistics (e.g. counting true positive (TP) and false positive (FP) variants) for each supercluster can be calculated using the resulting alignments. Unlike prior work, vcfdist attributes partial credit to variant calls which are not entirely correct yet still reduce the overall edit distance. We call these "partial positives" (PP). These partial positives can be used to more accurately represent the true performance of a variant caller, since it is common to report the nearly correct length of INDEL variants (such as in Fig. 1). The first step is identifying "sync points", where the optimal alignment path

passes through a reference base on the main diagonal (i.e. a base that is not within either a truth or query variant). Such sync points are marked in red in Supplementary Fig. 8d. Between each sync point, query and truth variants are considered to be true positives if the new alignment's edit distance is zero, false positives (or false negatives for the truth VCF) where the edit distance is unchanged, and partial positives where the edit distance is reduced. For the purposes of precision and recall calculation, partial positives are converted to a fractional positive plus a fractional negative using the old and new edit distances. For example, the partial positive query variant call in Fig. 1 reduces the edit distance from 3 to 1 and is counted as $\frac{2}{3}$TP + $\frac{1}{3}$FP. All statistics are calculated separately for each haplotype, and then summed. As a result, we count a false positive homozygous variant (1|1) as two false positives.

## Distance calculations

When measuring the distance between two sequences, we find it useful to define two metrics: "edit distance" (ED) and "distinct edits" (DE). For example, an insertion of length 5 has an edit distance of 5, but is 1 distinct edit. We would like to minimize both DE and ED simultaneously, placing greater importance on distinct edits than edit distance, since erroneously lengthening a gap is less detrimental than introducing new edits. If we aim to globally minimize 2DE + ED, this minimization problem is equivalent to Smith-Waterman global alignment with the affine-gap parameters $(m, x, o, e) = (0, 3, 2, 1)$. This is because opening a gap ($o$) increases DE, extending a gap ($e$) increases ED, and a substitution ($x$) increases both ED and DE. This affine-gap design point, which is point $B$ in Fig. 2a, is first used to align the query and truth sequences. Afterwards, the resulting alignment path is used to derive the edit distance and number of distinct edits. Note that although in other contexts "edit distance" colloquially refers to minimum edit distance, that is not the case here because design point $B$ attempts to simultaneously minimize DE and may do so at the expense of ED.

## New summary metrics

For each complex variant or INDEL, we allow partial positive variants to receive partial credit by comparing the reference sequence edit distance ($ED_{ref}$) and the query sequence edit distance ($ED_{query}$). This query sequence is derived by applying query VCF variants to the reference sequence.

Partial positives:

$$PP_{query} = \left(1 - \frac{ED_{query}}{ED_{ref}}\right)TP_{query} + \frac{ED_{query}}{ED_{ref}}FP_{query} \tag{1}$$

$$PP_{truth} = \left(1 - \frac{ED_{query}}{ED_{ref}}\right)TP_{truth} + \frac{ED_{query}}{ED_{ref}}FN_{truth} \tag{2}$$

In order to increase stability of precision calculations, vcfdist uses the number of true positives from the truth VCF rather than the query VCF.

GA4GH precision definition:

$$Precision = \frac{TP_{query}}{TP_{query} + FP_{query}} \tag{3}$$

vcfdist precision definition:

$$Precision = \frac{TP_{truth}}{TP_{truth} + FP_{query}} \tag{4}$$

In order to evaluate approximate genome variant calling quality, we define the F1 Q-score metric as follows:

$$F1_{Qscore} = -10 \log_{10}(1 - F1_{score}) \tag{5}$$

We make similar definitions for ED Q-score, DE Q-score, and ALN Q-score. These metrics are Q-score estimates of overall variant calling quality, based on the remaining edit distance, distinct edits, and alignment score of the query to truth sequences, relative to the difference between the reference and truth sequences:

$$ED_{Qscore} = -10 \log_{10}\left(\frac{ED_{query}}{ED_{ref}}\right) \tag{6}$$

$$DE_{Qscore} = -10 \log_{10}\left(\frac{DE_{query}}{DE_{ref}}\right) \tag{7}$$

$$ALN_{Qscore} = -10 \log_{10}\left(\frac{Alignment\ Score_{query}}{Alignment\ Score_{ref}}\right) \tag{8}$$

Lastly, we define the Average Maximum Rank Change (AMRC) to evaluate the stability of a dataset's relative performance with different variant representations. Stable VCF comparison methodologies are critical for determining promising directions for future research in variant calling, for curating clinical mutation databases, and for selecting current state-of-the-art variant calling pipelines. In this work, we evaluate the Q-score performance of pFDA Truth Challenge V2 submissions $S = \{S^{(1)}, ..., S^{(i)}, ..., S^{(n)}\}$. For each submission $S^{(i)}$, we use both the original ($O$) variant representations and when normalized to points $A$, $B$, $C$, and $D$.

$$S^{(i)} = \{S_O^{(i)}, S_A^{(i)}, S_B^{(i)}, S_C^{(i)}, S_D^{(i)}\} \tag{9}$$

First, we note the Q-score performance metrics for each submission $S^{(i)}$:

$$Q^{(i)} = Q_{score}(S^{(i)}) = \\ \{Q_{score}(S_O^{(i)}), Q_{score}(S_A^{(i)}), Q_{score}(S_B^{(i)}), Q_{score}(S_C^{(i)}), Q_{score}(S_D^{(i)})\} \tag{10}$$

Next, we create a sorted sequence (in increasing order) of the median Q-score for each submission, our best guess at each submission's true performance.

$$M = sort(\{m^{(i)} | m^{(i)} = med(Q^{(i)})\}) \tag{11}$$

Then we find the Q-scores for the worst- and best-performing representations of each submission.

$$W = \{w^{(i)} | w^{(i)} = min(Q^{(i)})\} \tag{12}$$

$$B = \{b^{(i)} | b^{(i)} = max(Q^{(i)})\} \tag{13}$$

Lastly, we find the average change in rank for submission $S^{(i)}$ when ranking submissions in order of Q-score if we use the best versus the worst possible representation. Note that we must omit the median of the current submission, med($Q^{(i)}$), when calculating the rank. We define pos($x$, $Y$) as the first position that value $x$ could be inserted into sorted sequence $Y$ such that it remains sorted.

$$AMRC = \sum_{i=1}^{n} pos(B^{(i)}, M - med(Q^{(i)})) - pos(W^{(i)}, M - med(Q^{(i)}))/n \tag{14}$$

## Global phasing calculation

To phase superclusters, we use a simple dynamic programming algorithm which minimizes the total number of switch errors and

supercluster phasing errors. A detailed example is shown in Supplementary Fig. 9. Firstly, superclusters are first categorized as type *X* or *Y* depending on the best mapping of query to truth haplotypes (*X*: Truth1 to Query1 and Truth2 to Query2, or *Y*: Truth1 to Query2 and Truth2 to Query1). It is also possible for both mappings result in the same minimum edit distance, in which case the supercluster is not categorized. During the forwards pass, switch errors and supercluster phasing errors are minimized, as the global phase state switches back and forth between *X* and *Y*. During the backwards pass, phase blocks are recovered and the total number of switch errors and supercluster phasing errors is reported.

### Output
In addition to a concise summary output (shown in Supplementary Fig. 4), vcfdist v1.0.0[46] outputs multiple verbose TSV files to aid in further analysis of variant calling performance. vcfdist also outputs both the original and new standard representations of the query and truth VCFs, as well as a GA4GH-compatible summary VCF that can be further analyzed using the helper script `qfy.py` from hap.py v0.3.15[47]. Lastly, several Python analysis scripts are provided which can be used to conveniently display the output of vcfdist.

### Evaluation
To evaluate the phased variant callsets, vcfdist v1.0.0[46] and the baseline rtg-tools vcfeval v3.12.1[9] were used. The full set of command-line parameters used for all analysis scripts can be found in the `analysis/` directory of the Github repository (https://github.com/TimD1/vcfdist).

### Reporting summary
Further information on research design is available in the Nature Portfolio Reporting Summary linked to this article.

## Data availability
All datasets used in this manuscript were previously released by other researchers and are publicly available online. The unphased precisionFDA Truth Challenge V2 VCF submissions and corresponding read FASTAs used to phase the VCFs are hosted on the NIST public data repository at https://data.nist.gov/od/id/mds2-2336(https://doi.org/10.18434/mds2-2336)[13]. The phased ground truth VCF and BED files for the HG002 whole-genome and challenging medically relevant genes datasets are available in the `NISTv4.2.1` and `CMRG_v1.00` directories of the Genome In A Bottle Consortium's FTP release folder, respectively: https://ftp-trace.ncbi.nlm.nih.gov/ReferenceSamples/giab/release/AshkenazimTrio/HG002_NA24385_son[19,35]. The GRCh38 reference FASTA is likewise available at https://ftp-trace.ncbi.nlm.nih.gov/ReferenceSamples/giab/release/references/. Source data are provided with this paper, and have been deposited in Zenodo in the `data/` folder under https://doi.org/10.5281/zenodo.8368282[46]. The file `Source Data.xlsx` is an Excel document of 16 sheets in total, with each sheet containing the raw data for each subfigure plot (Figs. 3a, b, c, 4a, b, c, d, 5bi, bii, biii, 6a, b, c, and Supplementary Fig. 5a–c). Each sheet contains a table listing the evaluation dataset, variant type and representation, submission ID, and evaluation metrics (precision, recall, edit distance, distinct edits, or F1 score) for each data point in the corresponding plot. Source data are provided with this paper.

## Code availability
All code for vcfdist and the benchmarking pipelines developed for this manuscript are available in a public Github repository (https://github.com/TimD1/vcfdist) under a permissive GNU GPLv3 license. It has also been deposited in Zenodo, under https://doi.org/10.5281/zenodo.8368282[46].

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

## Acknowledgements

This material is based upon work supported by the National Science Foundation Graduate Research Fellowship under Grants No. 1841052 (T.D.) and 2030454 (S.N.). Any opinion, findings, and conclusions or recommendations expressed in this material are those of the authors and do not necessarily reflect the view of the National Science Foundations. This work was also supported by the Kahn Foundation (S. N.).

## Author contributions

T.D. performed the computational studies. S.N. supervised the work.

## Competing interests

The authors declare no competing interests.
