## [Peer Review File · Nature Communications]

vcfdist: Accurately benchmarking phased small variant calls in human genomesREVIEWER COMMENTS

Reviewer #1 (Remarks to the Author):

In their manuscript, Dunn and Narayanasamy describe software for the accurate benchmarking of phased small variant calls in human genomes. Their software deals with the challenge of evaluating congruent variant representations in variant truth set evaluation. They highlight some known, long-standing challenges in the field of variation representation, and provide software that uses an intuitive sequence distance approach in lieu of conventional difference-based (“variant”) representations.

I found the results of this study to be highly engaging and an important work in the field of phased variant representation. The authors clearly convey the impact of variant caller parameterization on small substitution and indel variants, and provide several solutions for improving upon the current state-of-the-art tool (vcfeval) for benchmarking variant calls. The primary (and deliberate) limitation of their work—a requirement for input variant calls to provide local phasing annotations—will likely have the effect of encouraging variant callers to more consistently produce phased variant calls. I thought the authors addressed this rather well (if indirectly) by highlighting how vcfdist can be used as a preprocessing step for input to vcfeval; providing a path for alternate preprocessing steps that are not phasing-data dependent for historic datasets (or new datasets using technologies with limited phasing information).

This manuscript clearly demonstrates several key insights into the role of variant representation consistency in benchmarking, and makes a strong case for the evaluation of benchmarks using distance criteria over traditional variant match characteristics (even when compared with their introduced partial match characteristic).

I would like to commend the authors for adhering to core principles of open-source software development: a publicly-maintained repository with extensive commit history and a standard and permissible license (in this case, GPLv3).

I recommend that the authors undertake the following minor revisions to improve upon this already excellent work:

My primary point of confusion was an early reference to Figure 2 in the paper introduction. I would recommend not referencing this before the Result section, and to spend some time clarifying Figure 2 panel A. I went to reference the figure before making it to the Results and was immediately lost as to what I was looking at; the figure relies heavily on m, x, o, and e, which are first referenced in the figure caption, but may be better suited as additional in-panel legend material. Other elements of the panel are

left to guesswork without additional context. For example, the color scheme is not obvious to this reviewer, nor (initially) was the distinction between individual points, points connected by lines, and lines that do not connect points. It was not immediately intuitive why the preference gap axes are aligned as they are.

On reading through the Results section, much of this is clarified, e.g. lines not connecting points are not highlighted as they naïvely appear to be, but instead have “continuous” points. Some figure elements (specifically, the color scheme) remained difficult to decipher even after reading the entire work.

I advise the authors to check their latex source prior to resubmission; the rendered .pdf line numbers do not align with the written text, reducing their utility.

Regarding the use of the word “clever” near line index 68; while it is clearly meant to soften a critical comparison to the vcfeval application, this term is subjective and I recommend removing it or replacing it with a term that describes in what way the authors find it clever that can be quantified: “innovative”, “novel”, “efficient”, etc.

“This process of variant normalization” between line indices 79/80 is one of several standard approaches to normalizing small variants; other approaches include right-shifting (e.g. HGVS) and overprecision correction (e.g. NCBI SPDI, GA4GH VRS). I recommend expanding this background text to (minimally) reference these other approaches.

Reviewer #2 (Remarks to the Author):

The manuscript by Dunn et al. presents a new tool and approach for VCF variant call set benchmarking for phased variants, vcfdist. The challenge in variant call benchmarking arises from the representation of alternative variants in the VCF format, which describes variants in relation to a reference genome. This is particularly problematic for indels or multiple nucleotide variants. The authors identify several issues with the state-of-the-art tool, vcfeval, specifically with complex variant representation cases that may occur with long reads.

The major innovation introduced in this manuscript is the reinterpretation of the problem as a global pairwise alignment issue between the query and the reference. This insightful approach enables the resolution of complex variants, leveraging the local phasing provided by long reads and facilitating the

development of novel performance metrics. Furthermore, they have designed a "partial credit" system to account for partial matches between a complex call in the query and the truth set.

The vcfdist software showcased in this manuscript introduces a number of advancements and will prove to be a useful tool, particularly for long-read technology where local phasing is available. However, there are several questions that arise from this manuscript:

1. The authors imply throughout the manuscript that the shift towards long reads and the readily availability of local phasing are established facts. However, in reality, short reads continue to be the backbone of sequencing, particularly in the clinical setting, due to their cost-effectiveness. It's critical to acknowledge that while we may witness a transition to long reads in the next 5-10 years, for now, long reads remain a niche product. Therefore, a limitation of the authors' approach is the necessity for phased calls and this should not be minimized throughout the text.

2. The authors utilize the precision FDA v2 data and truth set to evaluate their tool. As they note, however, most of the data was unphased, and the source data from the Illumina technology was also unphased and did not provide local phasing. To remedy this, the authors employed computational phasing. However, the impact of phasing errors on the analysis remains unclear, even though the authors suggest it is minimal. Can they expand on the potential pitfalls of this approach? It might be beneficial for them to conduct an analysis exclusively with calls derived from ONT or PacBio platforms to demonstrate the potential performance achievable with these platforms.

3. The idea of "partial credit" is appealing and can offer valuable insights when comparing multiple calling methods. However, in a clinical benchmarking context, a partial call is considered an incorrect call and thus falls under false negatives (FN). The concept of partial credit could potentially mislead when evaluating the performance of a clinical pipeline in achieving specific sensitivity/specificity standards. Can this function be deactivated? I believe it would be helpful for the manuscript to include some comments on how to use this feature depending on the use case.

4. Does vcfdist provide ROC curves? While PR curves are useful for assessing overall performance, ROC curves are often more beneficial when comparing scoring systems and deciding on score thresholds, enabling users to set cutoffs for variant calls depending on the application. For instance, a research application might allow for more lenient FDR compared to a clinical application.

vcfdist: Response to Reviewers

Tim Dunn, Satish Narayanasamy

September 2023

Firstly, we would like to thank all reviewers for their time and effort in reviewing our paper. Their feedback and suggestions have been incredibly helpful towards improving our manuscript.

Responses to Reviewer #1:

In their manuscript, Dunn and Narayanasamy describe software for the accurate benchmarking of phased small variant calls in human genomes. Their software deals with the challenge of evaluating congruent variant representations in variant truth set evaluation. They highlight some known, long-standing challenges in the field of variation representation, and provide software that uses an intuitive sequence distance approach in lieu of conventional difference-based (“variant”) representations.

I found the results of this study to be highly engaging and an important work in the field of phased variant representation. The authors clearly convey the impact of variant caller parameterization on small substitution and indel variants, and provide several solutions for improving upon the current state-of-the-art tool (vcfeval) for benchmarking variant calls. The primary (and deliberate) limitation of their work—a requirement for input variant calls to provide local phasing annotations—will likely have the effect of encouraging variant callers to more consistently produce phased variant calls. I thought the authors addressed this rather well (if indirectly) by highlighting how vcfdist can be used as a preprocessing step for input to vcfeval; providing a path for alternate preprocessing steps that are not phasing-data dependent for historic datasets (or new datasets using technologies with limited phasing information).

This manuscript clearly demonstrates several key insights into the role of variant representation consistency in benchmarking, and makes a strong case for the evaluation of benchmarks using distance criteria over traditional variant match characteristics (even when compared with their introduced partial match characteristic).

I would like to commend the authors for adhering to core principles of open-source software development: a publicly-maintained repository with extensive commit history and a standard and permissible license (in this case, GPLv3).

I recommend that the authors undertake the following minor revisions to improve upon this already excellent work:

- 1. My primary point of confusion was an early reference to Figure 2 in the paper introduction. I would recommend not referencing this before the Result section, and to spend some time clarifying Figure 2 panel A. I went to reference the figure before making it to the Results and was immediately lost as to what I was looking at; the figure relies heavily on m , x , o , and e , which are first referenced in the figure caption, but may be better suited as additional in-panel legend material. Other elements of the panel are left to guesswork without additional context. For example, the color scheme is not obvious to this reviewer, nor (initially) was the distinction between individual points, points connected by lines, and lines that do not connect points. It was not immediately intuitive why the preference gap axes are aligned as they are.*

On reading through the Results section, much of this is clarified, e.g. lines not connecting points are not highlighted as they naïvely appear to be, but instead have “continuous” points. Some figure elements (specifically, the color scheme) remained difficult to decipher even after reading the entire work.

We have removed the early references to Figure 2 that were in the paper’s introduction. The parameters (m, x, o, e) have been added as a secondary legend as recommended. We modified Figure 2 to minimize potential confusions due to color (we removed the shaded regions and fixed a bug where the Python color scheme ran out of unique colors). The set of alignment parameters for each tool are now plotted in a unique color, except for where two tools have the exact same parameters. The preference axes were improved as well. Lastly, we have moved clarifying information from the Results section into the Figure description itself so that it can more easily be interpreted on its own.

2. I advise the authors to check their latex source prior to resubmission; the rendered PDF line numbers do not align with the written text, reducing their utility.

We have removed spacing between paragraphs and ensured a complete line skip between subsections. Nevertheless, the line numbers still do not align correctly on pages with figures or equations. We have used the *Springer Nature* L^AT_EX template, and do not believe this is a major issue since line numbers will be removed prior to publication. We apologize for any inconvenience this may cause during review.

3. Regarding the use of the word “clever” near line index 68; while it is clearly meant to soften a critical comparison to the vcfeval application, this term is subjective and I recommend removing it or replacing it with a term that describes in what way the authors find it clever that can be quantified: “innovative”, “novel”, “efficient”, etc.

We replaced the word “clever” with “innovative” on line 67.

4. “This process of variant normalization” between line indices 79/80 is one of several standard approaches to normalizing small variants; other approaches include right-shifting (e.g. HGVS) and overprecision correction (e.g. NCBI SPDI, GA4GH VRS). I recommend expanding this background text to (minimally) reference these other approaches.

Originally, we did not reference these other normalization methods because our manuscript focuses solely on variant normalization in the context of VCF files. However, we agree that this additional background and context may be useful to readers. We added a paragraph discussing competing standards for variant normalization in the contexts of general variant nomenclature (HGVS) and harmonized variant databases (SPDI, VRS) on lines 82-91.

Responses to Reviewer #2:

The manuscript by Dunn et al. presents a new tool and approach for VCF variant call set benchmarking for phased variants, vcfdist. The challenge in variant call benchmarking arises from the representation of alternative variants in the VCF format, which describes variants in relation to a reference genome. This is particularly problematic for indels or multiple nucleotide variants. The authors identify several issues with the state-of-the-art tool, vcfeval, specifically with complex variant representation cases that may occur with long reads.

The major innovation introduced in this manuscript is the reinterpretation of the problem as a global pairwise alignment issue between the query and the reference. This insightful approach enables the resolution of complex variants, leveraging the local phasing provided by long reads and facilitating the development of novel performance metrics. Furthermore, they have designed a “partial credit” system to account for partial matches between a complex call in the query and the truth set.

The vcfdist software showcased in this manuscript introduces a number of advancements and will prove to be a useful tool, particularly for long-read technology where local phasing is available. However, there are several questions that arise from this manuscript:

1. The authors imply throughout the manuscript that the shift towards long reads and the readily availability of local phasing are established facts. However, in reality, short reads continue to be the backbone of sequencing, particularly in the clinical setting, due to their cost-effectiveness. It’s critical to acknowledge that while we may witness a transition to long reads in the next 5-10 years, for now, long reads remain a niche product. Therefore, a limitation of the authors’ approach is the necessity for phased calls and this should not be minimized throughout the text.

For state-of-the-art WGS research pipelines, the shift to long reads as the backbone of sequencing has already occurred. This is evidenced by recent work from the Human Pangenome Reference Consortium (HPRC) [3], the Genome in a Bottle (GIAB) Consortium [10], the Human Genome Structural Variation Consortium (HGSVC) [7], and the Telomere-to-Telomere (T2T) Consortium [5, 8] – which recently finished the first truly complete human genome with the help of long reads. Without long reads, variant phasing and resolving long repetitive or repeated regions of the genome is impossible.

The purpose of `vcfdist`, as a benchmarking tool, is to accurately assess the relative performance of different long-read variant calling pipelines and technologies in the research and development phase. Because all the best-performing pipelines include long read sequencing, we can assume variants are locally phased without it limiting `vcfdist`'s applicability in this domain. As you correctly note, these long-read variant calling methodologies may take several years to mature and become cheaper before they are the preferred sequencing method in clinical practice.

For clinical WGS pipelines, we agree that short reads are the dominant sequencing modality due to the stability of the technology and its cost effectiveness in comparison to long reads. We do not tailor `vcfdist` towards unphased variants because the performance of short-read clinical pipelines is already well-understood, and there already exist tools designed for their evaluation such as `vcfeval` [1].

We have modified the *Discussion* section of our manuscript (lines 654-665) to include more context on the current state of WGS research pipelines and WGS in a clinical setting. While we believe the phasing requirement is not a limitation in terms of benchmarking long-read research pipelines, we do agree that it would be helpful for `vcfdist` to be able to compare unphased variants, which would allow it to be used in other contexts as well. We are currently working on extending `vcfdist` to evaluate unphased variants.

2a. The authors utilize the precision FDA v2 data and truth set to evaluate their tool. As they note, however, most of the data was unphased, and the source data from the Illumina technology was also unphased and did not provide local phasing. To remedy this, the authors employed computational phasing.

Clarification of our phasing methodology: most pFDA submissions provided unphased VCFs. The few submissions that were phased, we considered to be unphased. All original read data (HiFi, Illumina, and ONT) was also unphased. With such high coverage long read data (35× HiFi and 50× ONT), we were able to obtain high-quality phasings of the long reads, the short reads, and all the submitted VCFs. This was done using an existing tool, WhatsHap [4].

2b. However, the impact of phasing errors on the analysis remains unclear, even though the authors suggest it is minimal. Can they expand on the potential pitfalls of this approach?

We have added a discussion of the phasing error rate and why it does not impact our analysis to the paper's *Results* section (lines 578-608), and discuss our reasoning below.

There are two possible types of phasing errors: “flip” errors (in which a single variant is assigned the incorrect haplotype) and **“switch” errors** (in which many adjacent variants are assigned to the incorrect haplotype).

The rate of phasing errors is negligibly low. `vcfdist` is able to measure the switch error rate directly using the phasing analysis algorithm depicted in Supplementary Figure 9. `vcfdist` detected 1,337 switch errors for submission `K4GT3`, evaluated on the NIST dataset, which includes 5,423,813 variants. This amounts to one switch error every 4,057 variants (0.025%), or every 2.3 Mb.

Unfortunately, `vcfdist` is unable to directly measure the flip error rate. However, we can use the accuracy of the best-performing submissions to get an upper bound. Since the maximum SNP F1 score measured by `vcfdist` on the top few pFDA submissions is above Q35 (99.97% accuracy), the phasing error rate of individual variants is at most 0.03%.

Switch errors do not affect our analysis. Because `vcfdist` only enforces local and not global phasing, switch errors are allowed to occur between superclusters without affecting `vcfdist`'s analysis whatsoever. Switch errors are measured and reported later, but do not affect the total counts of true positive, false positive, and false negative variants.

Flip errors impact accuracy, but not the conclusions of this manuscript. The primary goal of this manuscript is to demonstrate that `vcfdist` is less sensitive to variant representation than `vcfeval` as a result of our methodological contributions (Figures 3, 4, 6). Since phasing occurs upstream to evaluation

with `vcfdist`, flip errors could reduce the measured performance of each variant caller (with the same impact as if the original variant caller had phased variants incorrectly), but this does not impact any of our results in regard to evaluation stability. In other words, flip errors will slightly lower the values of the precision-recall curves in Figure 4d for each representation uniformly. The curves will remain similarly clustered, much closer together than when evaluated with `vcfeval`. Furthermore, because we used the same exact reads to phase all VCF submissions, we can expect the same low phasing error rate across all VCFs. This ensures their measured relative performance is unchanged.

2c. It might be beneficial for them to conduct an analysis exclusively with calls derived from ONT or PacBio platforms to demonstrate the potential performance achievable with these platforms.

Many of the pFDA submissions we evaluated actually used reads from only a subset of the three sequencing technologies (ONT, Illumina, and PacBio) for calling variants. We refer the reviewer to the original pFDA paper [6] which already contains a comparison of these sequencing technologies. A comparison of phasing accuracy specifically has also been done before, in [9].

3. The idea of “partial credit” is appealing and can offer valuable insights when comparing multiple calling methods. However, in a clinical benchmarking context, a partial call is considered an incorrect call and thus falls under false negatives (FN). The concept of partial credit could potentially mislead when evaluating the performance of a clinical pipeline in achieving specific sensitivity/specificity standards. Can this function be deactivated? I believe it would be helpful for the manuscript to include some comments on how to use this feature depending on the use case.

The intended application of `vcfdist` at the moment is general benchmarking, and therefore partial credit is enabled internally by default. For other use cases, `vcfdist` provides several verbose output TSVs and a summary VCF in the standard GA4GH output format [2]. For example, `precision-recall.tsv` includes the raw true positive, partial positive, false positive, and false negative variant counts for each variant type at every integer quality score. The desired metrics (such as precision and recall without partial credit) can easily be calculated using this data.

As recommended, we have added a short discussion in our *Results* section on when using partial credit would or would not make sense (lines 518-525). Additionally, we have augmented our `vcfdist` demo with a Python script that generates a precision/recall plot either with or without partial credit, as specified by a boolean flag.

4. Does `vcfdist` provide ROC curves? While PR curves are useful for assessing overall performance, ROC curves are often more beneficial when comparing scoring systems and deciding on score thresholds, enabling users to set cutoffs for variant calls depending on the application. For instance, a research application might allow for more lenient FDR compared to a clinical application.

Although ROC curves are useful for assessing clinical decision-making pipelines, they are less applicable in the context of whole-genome sequencing (WGS) benchmarking. For clinical diagnostics, there is often a list of known pathogenic variants that are being tested for. For whole-genome sequencing, however, there are infinitely many possible variants that could be called (substitutions, deletions, or insertions of any length at any position). As a result, there are an infinite number of “true negative” variant calls and there is no meaningful calculation of FPR for use in a ROC curve. We simply provide total counts of TP/FP/FN/PP variant calls and allow the user to calculate additional metrics such as FDR if they desire.

References

- [1] John G Cleary et al. “Comparing variant call files for performance benchmarking of next-generation sequencing variant calling pipelines”. In: *BioRxiv* (2015), p. 023754.
- [2] Peter Krusche et al. “Best practices for benchmarking germline small-variant calls in human genomes”. In: *Nature biotechnology* 37.5 (2019), pp. 555–560.
- [3] Wen-Wei Liao et al. “A draft human pangenome reference”. In: *Nature* 617.7960 (2023), pp. 312–324.

- [4] Marcel Martin et al. “WhatsHap: fast and accurate read-based phasing”. In: *BioRxiv* (2016), p. 085050.
- [5] Sergey Nurk et al. “The complete sequence of a human genome”. In: *Science* 376.6588 (2022), pp. 44–53.
- [6] Nathan D Olson et al. “PrecisionFDA Truth Challenge V2: Calling variants from short and long reads in difficult-to-map regions”. In: *Cell Genomics* 2.5 (2022), p. 100129.
- [7] Mikko Rautiainen et al. “Telomere-to-telomere assembly of diploid chromosomes with Verkko”. In: *Nature Biotechnology* (2023), pp. 1–9.
- [8] Arang Rhie et al. “The complete sequence of a human Y chromosome”. In: *Nature* (2023), pp. 1–11.
- [9] Kishwar Shafin et al. “Haplotype-aware variant calling with PEPPER-Margin-DeepVariant enables high accuracy in nanopore long-reads”. In: *Nature methods* 18.11 (2021), pp. 1322–1332.
- [10] Justin M Zook et al. “A robust benchmark for detection of germline large deletions and insertions”. In: *Nature biotechnology* 38.11 (2020), pp. 1347–1355.

REVIEWERS' COMMENTS

Reviewer #1 (Remarks to the Author):

The author response to my recommended revisions is satisfactory.

Reviewer #2 (Remarks to the Author):

I would like to thank the authors for comprehensively responding to the queries from the reviewers. Upon reading their response, I'm convinced that all outstanding queries have been addressed, and the changes made to the manuscript have enhanced the report.

The only response with which I disagree pertains to the query about ROC curves. ROC curves are indeed perfectly applicable for benchmarking WGS call sets against a truth set based on variant scores, and this is routinely done. Perhaps I should have been more explicit about the type of ROC curve I had in mind: sensitivity (TP rate) versus FP counts (or precision) by thresholds of score values. Such curves can be constructed from the output of `vcfeval` using the tool `vcfplot` for WGS data. In the context of WGS for clinical applications, precision is typically of greater concern than specificity, especially given the reasons the authors pointed out about the infinite number of false negatives. These types of ROC curves are crucial for deciding on thresholds of acceptable trade-offs between sensitivity and precision, and for identifying cut-offs for practical variant calling pipelines. Thus, I maintain that it would be beneficial for `vcfdist` to generate tables that can be plotted as these curves, much like the tabular output from `vcfeval`.

Nevertheless, even without this feature, I believe the manuscript is ready for publication.

vcfdist: Response to Reviewers' Responses

Tim Dunn, Satish Narayanasamy

November 2023

We would like to thank all reviewers for their time and effort in reviewing our responses.

Responses to Reviewer #1:

The author response to my recommended revisions is satisfactory.

Thank you.

Responses to Reviewer #2:

I would like to thank the authors for comprehensively responding to the queries from the reviewers. Upon reading their response, I'm convinced that all outstanding queries have been addressed, and the changes made to the manuscript have enhanced the report.

The only response with which I disagree pertains to the query about ROC curves. ROC curves are indeed perfectly applicable for benchmarking WGS call sets against a truth set based on variant scores, and this is routinely done. Perhaps I should have been more explicit about the type of ROC curve I had in mind: sensitivity (TP rate) versus FP counts (or precision) by thresholds of score values. Such curves can be constructed from the output of vcfeval using the tool vcfplot for WGS data. In the context of WGS for clinical applications, precision is typically of greater concern than specificity, especially given the reasons the authors pointed out about the infinite number of false negatives. These types of ROC curves are crucial for deciding on thresholds of acceptable trade-offs between sensitivity and precision, and for identifying cut-offs for practical variant calling pipelines. Thus, I maintain that it would be beneficial for vcfdist to generate tables that can be plotted as these curves, much like the tabular output from vcfeval.

Nevertheless, even without this feature, I believe the manuscript is ready for publication.

I'm glad that we were able to successfully address most of your queries. In regards to ROC curves, it seems that there has been a miscommunication: **our tool vcfdist does already provide exactly what you are asking for.**

Above, you state that “Perhaps I should have been more explicit about the type of ROC curve I had in mind: sensitivity (TP rate) versus FP counts (or precision) by thresholds of score values.” This is what we define as a “precision-recall” curve: sweeping the thresholds of variant quality score values and reporting the sensitivity (also known as “recall”) versus precision.

You then state “Thus, I maintain that it would be beneficial for vcfdist to generate tables that can be plotted as these curves, much like the tabular output from vcfeval”. We actually modeled vcfdist's tabular output directly after vcfeval, and provide many of the same outputs for each quality score threshold, including (but not limited to) precision, recall, F1 score, total variants, false positives, true positives, and false negatives.